# Euvichol-plus vaccine campaign coverage during the 2017/2018 cholera outbreak in Lusaka district, Zambia: a cross-sectional descriptive study

Victor M Mukonka,[1,2] Cephas Sialubanje [iD],[1] Belem Blamwell Matapo,[3] Orbrie Chewe,[4,5] Albertina Moraes Ngomah [iD],[6] Willaim Ngosa,[6] Raymond Hamoonga,[4] Nyambe Sinyange,[4] Hannah Mzyece,[4] Lucy Mazyanga,[6] Nathan Bakyaita,[7] Nathan Kapata[8]

For numbered affiliations see end of article.

**Correspondence to**
Dr Cephas Sialubanje;
csialubanje@yahoo.com

## ABSTRACT

**Objective** To determine the coverage for the oral cholera vaccine (OCV) campaign conducted during the 2017/2018 cholera outbreak in Lusaka, Zambia.

**Study design** A descriptive cross-sectional study employing survey method conducted among 1691 respondents from 369 households following the second round of the 2018 OCV campaign.

**Study setting** Four primary healthcare facilities and their catchment areas in Lusaka city (Kanyama, Chawama, Chipata and Matero subdistricts).

**Participants** A total of 1691 respondents 12 months and older sampled from 369 households where the campaign was conducted. A satellite map-based sampling technique was used to randomly select households.

**Data management and analysis** A pretested electronic questionnaire uploaded on an electronic tablet (ODK V.1.12.2) was used for data collection. Descriptive statistics were computed to summarise respondents' characteristics and OCV coverage per dose. Bivariate analysis ($\chi^2$ test) was conducted to stratify OCV coverage according to age and sex for each round (p<0.05).

**Results** The overall coverage for the first, second and two doses were 81.3% (95% CI 79.24% to 83.36%), 72.1% (95% CI 69.58% to 74.62%) and 66% (95% CI 63.22% to 68.78%), respectively. The drop-out rate was 18.8% (95% CI 14.51% to 23.09%). Of the 81.3% who received the first dose, 58.8% were female. Among those who received the second dose, the majority (61.0%) were females aged between 5 and 14 years (42.6%) and 15 and 35 years (27.7%). Only 15.5% of the participants aged between 36 and 65 and 2.5% among those aged above 65 years received the second dose.

**Conclusion** These findings confirm the 2018 OCV campaign coverage and highlight the need for follow-up surveys to validate administrative coverage estimates using population-based methods. Reliance on health facility data alone may mask low coverage and prevent measures to improve programming. Future public health interventions should consider sociodemographic factors in order to achieve optimal vaccine coverage.

## STRENGTHS AND LIMITATIONS OF THIS STUDY

⇒ Conducting the survey shortly (within a month) after the second round of the oral cholera vaccine campaign minimised recall bias.
⇒ Verification of the self-reported vaccination status with the vaccination certificate card increased internal validity of the findings.
⇒ Use of a satellite map-based sampling technique to randomly select households reduced selection bias.
⇒ Use of a pretested electronic questionnaire enhanced data quality and ensured internal validity of the study.
⇒ Although a random satellite-based sampling technique was used to select households, inclusion of all eligible participants found in a household into the study may have introduced selection bias.

## INTRODUCTION

### Background

Cholera is a rapidly dehydrating diarrhoeal illness caused by the toxigenic bacterium *Vibrio cholerae* serogroups O1 and O139.[1] The O1 serogroup has two biotypes, classical and E1 Tor.[2] During the past years, cholera has spread from the Indian subcontinent to involve nearly the whole world and cause pandemics in Africa and South America. Since 1817, the world has experienced seven cholera pandemics, the first six of which were caused by the classical O1 biotype. The current (seventh) pandemic, which began in 1961 and largely confined to Asia and Africa, is mainly caused by the El Tor O1 biotype which has essentially displaced the classical biotype globally since 1993.[2,3] Overcrowding, poverty, poor sanitation facilities, lack of access to clean and safe water and food contamination increase the risk for cholera outbreaks.[4] Global annual estimates of cholera range from 1.3 to 4. million cases and



21 000 to 143 000 deaths.[5–7] However, the actual burden of the disease is unknown due to poor reporting, limited surveillance and lack of laboratory capacity. In Africa, the burden of cholera is mainly in sub-Sahara Africa with approximately 19 countries more affected by epidemics, with 105 287 cases and 1882 deaths.[5–7]

Although access to and provision of safe drinking water, adequate sanitation, disease surveillance, health promotion and case management remain the mainstay of cholera prevention and control,[8] use of reasonably priced, safe and effective vaccines can provide an additional intervention for cholera prevention and control.[9] The 2011 World Health Assembly[10] resolution 64.15 called for implementation of an integrated and comprehensive approach to cholera control. Consequently, in 2011, the WHO established a global stockpile under the supervision of the International Coordination Group on Vaccine Provision to strengthen the capacity for action against cholera in emergency settings and recommended use of oral cholera vaccines (OCV) in endemic and epidemic settings.[11] Subsequently, four killed, whole-cell OCVs–Dukoral, mORCVAX, Shanchol and Euvichol–are currently available and licensed for use in different countries.[11] Shanchol and Euvichol are killed, whole-cell, bivalent O1 and O139 and licensed for use in children ≥1 year and require two doses given 2 weeks apart. They are the preferred vaccines for public health use as they do not need a buffer.[12]

Numerous studies have demonstrated the efficacy, safety and immunogenicity of the killed whole-cell OCV.[13–15] For example, a randomised placebo-controlled safety and immunogenicity study conducted among healthy adults in cholera endemic area in Kolkata, India[16] found that the reformulated oral killed whole cell vaccine was safe, well tolerated and highly immunogenic. Studies have also shown that Euvichol has the same formulation as Shanchol and that it has non-inferior seroconversion rates to those of Shanchol.[17 18] The study also reported a four-fold rise in serum *V. cholera* O1 vibriocidal antibody titers among 53% adults and 80% children.

Following the 2017/2018 cholera outbreaks[19] and based on lessons from the previous outbreaks,[20] the Zambian Ministry of Health (MoH) through WHO and with support from the Global Task Force on Cholera Control, procured 2 070 100 vaccine doses of the Euvichol-Plus OCV campaign to support the short-term to medium-term interventions to control cholera. The first OCV round was launched on the 10 January 2018 in the four affected subdistricts of Lusaka (ie, Chawama, Matero, Kanyama and Chipata). Due to logistical challenges, and while waiting for delivery of the complete vaccine stocks of OCV doses, the second round of the campaign was implemented in two phases. Phase 1 was conducted between 5 February 2018 and 14 February 2018 and covered two of the four subdistricts (Chawama and Kanyama); phase 2 was implemented from 18 April 2018 to 25 April 2018 and included the other two subdistricts (Chipata and Matero). In addition to the OCV campaign, the Zambian MOH implemented the following interventions: provision of safe water, chlorination of unsafe water sources, improved sanitation and solid waste management, training of community workers, cleaning of the environment with community involvement and health promotion campaigns through door-to-door campaigns, public address system and mass media (radio and television).

## Objectives

The aim of this study was to determine the actual OCV coverage during the 2018 OCV campaign conducted in Lusaka district. Information on the actual vaccination coverage (based on the population vaccinated) is required to validate the administrative coverage that was reported at 109% for the first round. Currently, there is a lack of evidence on the coverage of OCV campaigns among the Zambian population.

## METHODS
### Study design

This was a survey conducted among 1691 respondents from 369 households within a month following the second round of the 2018 OCV campaign in the four subdistricts of Lusaka, Zambia.

### Study site and participants

The study sites included the four subdistricts of Lusaka (Kanyama, Chawama, Chipata and Matero) where the 2018 OCV campaign was conducted. The combined population for the four subdistricts was 1543 507 (see our previous work on the 2018 Lusaka cholera outbreak and for the details on the study setting).[21–23] The study population included all individuals older than 12 months residing in the study sites.

### Inclusion criteria

To be eligible to participate in the survey, respondents needed to:
1. Be residing in study sites for at least 2 weeks prior to the last date of vaccine administration.
2. Older than 12 months.
3. Provide consent to participate in the study.

### Sample size determination

To estimate the sample size for the survey, the following assumptions were made: (a) 50% of the population received two doses; (b) CI of 95% (equivalent to an alpha error of 5%); precision of 5; (c) design effect of 3 and (d) 15% missing data and non-response. This gave an estimated sample size of 1, 325 of respondents. To estimate the required number of households, the Zambia Demographic and Health Survey report of 2013–2014,[24] which showed a mean number of four persons per household was used. This gave a total of 332 households (1325/4=331.25) to be included in the study. A household was defined as a group of people sleeping under the same roof and sharing meals every day for at least two previous weeks.

**Table 1** Demographic characteristics

| Variable | n (%) |
|---|---|
| **Sex** | |
| Male | 697 (41.2) |
| Female | 994 (58.8) |
| **Age (in years)** | |
| Below 5 | 185 (10.9) |
| 5–9 | 263 (15.6) |
| 10–19 | 453 (26.8) |
| 20–29 | 318 (18.8) |
| 30–39 | 213 (12.6) |
| 40–49 | 136 (8.0) |
| 50–59 | 59 (3.5) |
| 60 and above | 64 (3.8) |
| **Level of education** | |
| Never attended | 249 (14.7) |
| Primary | 738 (43.6) |
| Secondary | 614 (36.3) |
| Tertiary | 90 (5.3) |
| **Occupation of the household head (N=357)** | |
| Formal employment | 57 (16.0) |
| Unemployed | 144 (40.3) |
| Informal employment | 144 (40.3) |
| Student | 2 (0.6) |
| Other | 10 (2.8) |

## Sampling techniques

A random satellite-based sampling technique was used to select households for inclusion in the study. First, a random point was generated in the sampling area and retained if it coincided with a household (±10 m error); the process was repeated until the required sample size was reached. All eligible individuals (ranging from 1 to 6 participants per household) in the selected household at the time of the survey were selected.

## Data collection

Twenty-four trained and experienced research assistants were recruited from the previously conducted coverage survey in Lusaka district. Research assistants were trained for 4 days; the training comprised classroom-based theory for 3 days followed by field work for 1 day. Topics covered during classroom training included the (a) purpose of the study, (b) study methods including sampling techniques and (c) electronic questionnaire administration. Field work included testing the data collection tools, household location using geographical positioning system (GPS) technique. Data collection instruments were revised based on feedback from the research assistants.

Research assistants worked in teams of four, comprising three members and a team leader. Research assistants administered the questionnaire and read out the interview questions during the face-to-face interview which were conducted in the local language. In addition, the research assistants collected information on the number and proportion of people that received the vaccine in the household; collected GPS coordinates and data on health promotion and cholera prevention. If responders were not found at the house, a second visit was organised later in the day or on a second day. The household was skipped if respondents were not found or refused to participate during the second visit.

## Data collection instruments

A pretested electronic questionnaire uploaded on the electronic tablet using ODK V.1.12.2 was used to collect data during face-to-face interviews. The questionnaire comprised questions on eligibility, sociodemographic data (age, sex and household size), vaccination status (self-reported and card-confirmed) and acceptability data (perception and beliefs about the vaccine, vaccination adverse events). To ensure quality in the data collection process, the team leader cross checked the completed questionnaires on a daily basis. Completed and verified questionnaires were transmitted to the server at Zambia National Public Health Institute.

## Quality assurance

To ensure quality assurance in the data collection process, the following measures were taken: (a) the survey was conducted about 2 weeks after completion of the second round of the vaccination campaign to minimise recall bias; (b) research assistants were trained on data collection methods; (c) an electronic data collection tool uploaded on the ODL kit was pretested and revised accordingly; (d) field supervisors worked with the research assistants and verified the data entry before the data were transmitted to the sever; (e) research assistants had no access to the transmitted data and (f) householders who were not present during the field were revisited to avoid selection bias.

## Data processing and analysis

Data were retrieved from the server, cleaned and saved on an excel sheet. Analysis was done using STATA V.12.4. Descriptive statistics were computed to summarise respondent and household characteristics and OCV campaign coverage for each round. Vaccination status was determined using both the vaccination certificate (card-confirmed) and self-reports. In addition, $\chi^2$ test was used to compute the OCV vaccine coverage by sex and age group for each round (95% CI, p<0.005). To estimate coverage of the 2018 vaccination campaign, the following indicators were computed: (a) vaccine coverage for two doses, (b) vaccine coverage for the first round; (c) vaccine coverage for the second round and (d) vaccine coverage with at least one dose. Vaccine coverage was calculated by dividing the number of individuals reporting being vaccinated by the survey population and expressed as a

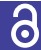

**Table 2** Household characteristics and risk factors (n=357)

| Variable | n (%) |
|---|---|
| Source of drinking water | |
| Borehole | 119 (33.3) |
| Bottled water/company for selling water | 4 (1.1) |
| Piped water (in the yard) | 67 (18.8) |
| Piped water (in the house) | 44 (12.3) |
| Piped water (public) | 107 (30.0) |
| Well (with a pump) | 12 (3.4) |
| Well (unprotected) | 3 (0.8) |
| Other | 1 (0.3) |
| Treatment of drinking water | |
| No | 113 (31.7) |
| Yes | 244 (68.3) |
| Methods used to treat water | |
| Adding chlorine | 134 (54.9) |
| Adding chlorine using water filter | 1 (0.4) |
| Boil water | 41 (16.8) |
| Boil water and adding chlorine | 63 (25.8) |
| Boil water and using water filter | 1 (0.4) |
| Letting water stand/stand | 1 (0.4) |
| Using water filter | 2 (0.8) |
| Other | 1 (0.4) |
| Use of flush toilet connected to system | |
| Yes | 58 (16.2) |
| No | 299 (83.8) |
| Use of flush toilet connected to septic tank | |
| Yes | 67 (18.2) |
| No | 290 (81.2) |
| Use of flush toilet connected to septic tank | |
| Yes | 11 (3.1) |
| No | 346 (96.9) |
| Use of pit latrines, ventilated or ameliorated | |
| Yes | 14 (3.9) |
| No | 343 (96.1) |
| Pit latrine with cement slab | |
| Yes | 168 (47.1) |
| No | 189 (52.9) |
| Pit latrine without cement slab | |
| Yes | 38 (10.6) |
| No | 319 (89.4) |
| No toilet (canal, open defecation, bush, field) | |
| Yes | 5 (1.4) |
| No | 352 (98.6) |
| Toilet facilities: others | |
| Yes | 3 (0.8) |
| No | 354 (99.2) |

Continued

**Table 2** Continued

| Variable | n (%) |
|---|---|
| Toilet facilities shared with other households | |
| Yes | 182 (51.0) |
| No | 175 (49.0) |
| Washing of hands during the day | |
| Yes | 349 (97.8) |
| No | 8 (2.2) |
| Washing of hands after using the toilet | |
| Yes | 349 (98.3) |
| No | 6 (1.7) |
| Washing hands before eating | |
| Yes | 319 (91.4) |
| No | 30 (8.6) |
| Washing hands after eating (n=319 who answered yes above) | |
| Yes | 268 (76.8) |
| No | 81 (23.2) |
| Washing hands before cooking | |
| Yes | 194 (55.6) |
| No | 155 (44.4) |
| Washing hands after cleaning tables | |
| Yes | 79 (22.6%) |
| No | 270 (77.4%) |
| Washing hands after cleaning baby stool/diapers | |
| Yes | 113 (32.4) |
| No | 236 (67.6) |
| Washing hands after cleaning house | |
| Yes | 120 (34.4) |
| No | 229 (65.6) |
| Washing hands any time | |
| Yes | 67 (19.2) |
| No | 282 (80.8) |
| Washing hands any other time | |
| Yes | 4 (1.1) |
| No | 345 (98.9) |

percentage. In addition, the drop-out rate the two rounds was calculated as shown below:

$$Drop\ out\ rate = \left( \frac{Vaccin\_1 - Vaccin\_1\&\_2}{Vaccin\_1} \times 100 \right)$$

Vaccin_1=the number of people vaccinated during the first round.

Vaccin_1& 2=the number of people vaccinated during the first and the second round.

## Patient and public involvement

The study design was determined by the research team. Participants and the public were not directly involved in

the conceptualisation and design of the study. However, selection of the study sites was done in consultation with stakeholders from the MoH and Zambia National Public Health Institute. Selection of study participants was done in collaboration with the provincial and district health managers. A dissemination meeting was held in Lusaka and study findings shared with key stakeholders, including the WHO, MoH and community leaders in the health facilities where the study was conducted. A final report was also written and shared with the funding organisation.

## RESULTS
### Participants
A total of 2000 participants were recruited into the study. Of these, 200 (10%) could not be traced due to access challenges in the urban slums and thus dropped out of the study. A total of 1800 (90%) participants completed the study, out of whom 109 (5.5%) had incomplete records and were not included in the final analysis. A total final sample of 1, 691 respondents (84.5%) were included in the final analysis. A summary of the recruitment algorithm of study participants is shown in online supplemental figure 1.

### Demographic characteristics
Most respondents were female (58.8%) and aged below 30 (53.3%). Slightly more than two-fifths (43.6%) of the respondents had completed primary education, 36.3% had completed secondary education and 5.3% had attended tertiary education and 14.7% never attended school. With regard to occupation, 16.0% were in formal employment, 40.3% were in informal employment and 40.3% were unemployed (table 1).

### Household characteristics and risk factors
The characteristics of the 357 households that were included in the study are shown in table 2. With regard to source of drinking water, 33.3% used borehole water, 30% used piped communal water, 18.8% used piped water from the yard and 12.3% used piped water from within the house and majority (68.3%) treated their drinking water. Of those who treated their drinking water, 54.9% added chlorine, 25.8% boiled water and added chlorine, 6.8% only boiled their water. Concerning use of toilets, only 16.2% used to flush toilets connected to the system, 18.2% used a flush toilet connected to a septic tank, 3.9% used ventilated pit latrines. Majority (98.3%) washed their hands after using the toilet and 91.4% washed their hands before eating (table 2).

### Analysis of respondents who received the first and second OCV doses (n=1, 691)
Out of the total sample, 1, 375 (81.3%) received the first dose. Of these, 994 (58.8%) were female, 185 (10.9%) were aged between 1 and 4 years, 637 (37.7%) between 5 and 7 years, 543 (32.1%) between 18 and 35 years, 285

**Table 3** Respondents who received the first and second OCV doses

| First OCV dose | | | |
|---|---|---|---|
| | **Yes (n=1, 375)** | **No (n=316)** | |
| **Variable** | **n (%)** | **n (%)** | **Total (n=1, 691)** |
| Sex | | | |
| Male | 556 (40.4) | 141 (20.2) | 697 (41.2) |
| Female | 819 (59.6) | 175 (17.6) | 994 (58.8 |
| Age (in years) | | | |
| 1–4 | 158 (11.5) | 27 (8.5) | 185 (10.9) |
| 5–14 | 554 (40.3) | 83 (26.3) | 637 (37.7) |
| 15–35 | 409 (29.7) | 134 (42.4) | 543 (32.1) |
| 36–65 | 226 (16.4) | 59 (18.7) | 285 (16.9) |
| Above 65 | 28 (2.1) | 13 (4.1) | 41 (2.4) |
| **Second dose** | | | |
| | **Yes (n=1, 220)** | **No (n=471)** | |
| **Variable** | **n (%)** | **n (%)** | **Total (n=1, 691)** |
| Sex | | | |
| Male | 476 (39.0) | 234 (49.6) | 710 (42.0) |
| Female | 744 (61.0) | 237 (50.4) | 981 (58.0) |
| Age | | | |
| 1–4 | 143 (11.7) | 46 (9.8) | 189 (11.1) |
| 5–14 | 520 (42.6) | 159 (33.8) | 679 (40.2) |
| 15–35 | 338 (27.7) | 172 (36.5) | 510 (30.2) |
| 36–65 | 189 (15.5) | 87 (18.4) | 276 (16.3) |
| Above 65 | 30 (2.5) | 7 (1.5) | 37 (2.2) |
| OCV, oral cholera vaccine. | | | |

(16.9%) were aged between 36 and 65; 41 (2.4%) were aged above 65 years. The differences in characteristics between the respondents who received and those who did not receive the OCV during the first round are shown in table 3.

### Analysis of respondents who received the second OCV doses (n=1, 691)
Out of the total sample of 1691 that was included in the analysis for the second round of the OCV campaign, 1, 220 (72.1%) received the second dose. Of these, 744 (61.0%) were female, 143 (11.7%) were aged between 1 and 4 years, 520 (42.6%) were between 5 and 14 years, 338 (27.7%) were between 15 and 35 years, 189 (15.5%) were aged between 36 and 65; 30 (2.5%) were aged above 65 years (table 4).

### Administrative coverage
The overall administrative coverage for the OCV first dose was 110%. Kanyama subdistrict reported the highest coverage at 141.9%, followed by Chipata at 124.1%, Chawama at 98.4%; Matero subdistrict had the lowest at 82.4%. The overall coverage for the second dose was 89%. Kanya subdistrict recorded the highest coverage at 158%,



**Table 4** Respondents who received the first and second OCV dose

| Second OCV dose | | | |
|---|---|---|---|
| Variable | Yes (n=1, 220) | No (n=471) | Total (n=1, 691) |
| | n (%) | n (%) | |
| Sex | | | |
| Male | 476 (39.0) | 234 (49.6) | 710 (42.0) |
| Female | 744 (61.0) | 237 (50.4) | 981 (58.0) |
| Age | | | |
| 1–4 | 143 (11.7) | 46 (9.8) | 189 (11.1) |
| 5–14 | 520 (42.6) | 159 (33.8) | 679 (40.2) |
| 15–35 | 338 (27.7) | 172 (36.5) | 510 (30.2) |
| 36–65 | 189 (15.5) | 87 (18.4) | 276 (16.3) |
| Above 65 | 30 (2.5) | 7 (1.5) | 37 (2.2) |

OCV, oral cholera vaccine.

followed by Chawama at 98%, Chipata at 63%. Matero recorded the lowest coverage at 62.9%

### OCV campaign coverage by dose based on the survey (n=1, 691)

Table 5 shows the OCV campaign coverage for each dose based on the survey results. The vaccine coverage for the first dose was 81.3% (95% CI 79.24% to 83.36%); the coverage for the second dose was 72.1% (95% CI 69.58% to 74.62%); the two-dose coverage was 66% (95% CI 63.22% to 68.78%). The drop-out rate was 18.8% (95% CI 14.51% to 23.09%) (table 5).

### DISCUSSION

The aim of this study was to determine the actual coverage (based on the population vaccinated) for the OCV campaign that was conducted in Lusaka district in 2018. Overall, our findings showed that the reported administrative coverage was much higher than the actual coverage. Coverage for the two doses was 66.0%; coverage for the first and second dose was 81.3% and 72.1%, respectively. The drop-out rate was 18.8%.

For both the first and second campaigns, the majority of the participants who received the vaccine were female in the age range 5 and 14 years (42.6%) and 15 and 35

**Table 5** OCV campaign coverage by dose

| OCV dose | No vaccinated | % (95% CI) |
|---|---|---|
| First dose | 1375 | 81.3 (79.24 to 83.36) |
| Second dose | 1220 | 72.1 (69.58 to 74.62) |
| Both doses | 1116 | 66 (63.22 to 68.78) |
| Drop-out rate | 318 | 18.8 (14.51 to 23.09) |
| At least one dose* | 363 | 21.5 (17.27 to 25.73) |

*Excludes those that received both doses.
OCV, oral cholera vaccine.

years (27.7%). Only 15.5% and 2.5% of the participants aged between 36 and 65 and above 65 years, respectively, received the second dose.

These findings show that the administrative coverage reported during the first and second rounds was much higher than the actual one from the population survey. These findings are consistent with a survey conducted by Semá Baltazar et al[25] in Mozambique which reported a difference between the OCV administrative coverage and the one from the population survey. According to these authors, the administrative coverage provided by the Mozambican MoH were 108.4% and 107.9% after the first and second rounds, respectively. However, based on the population survey, the overall vaccination coverage with at least one dose was 69.5% (95% CI 51.2% to 88.2%); the full dose coverage (two doses) was 51.2% (95% CI 37.9% to 64.3%). The discrepancy between the administrative and survey coverages could be due to data quality challenges and low population denominators. For example, in their study on monitoring vaccination coverage, Cutts et al[26] showed that the population denominator and the reported number of vaccinations used in administrative estimates are often inaccurate. These authors concluded that, because of the inaccuracy associated with administrative data, surveys are often considered to be more accurate. Reliance on health facility data alone for calculating the OCV campaign coverage may mask low coverage and prevent subsequent measures to improve programming. Thus, there is a need for periodic validation of administrative coverage estimates with population-based methods.[27]

Our findings provide evidence on the feasibility of conducting an OCV campaign during a cholera outbreak according to the WHO guidelines which require conducting OCV campaign during an outbreak in addition to traditional interventions. The findings also corroborate previous studies which reported OCV campaign coverage using population survey data. For example, a survey conducted by Bwire et al[28] on the use of surveys to evaluate an integrated OCV campaign in response to a cholera outbreak in Hoima district, Uganda, reported a (94% (95% CI 92% to 95%) coverage for at least one OCV dose and 78% (95% CI 76% to 81%) for two doses. Similarly, a Malawian survey by Sauvageot et al[29] reported a high two-dose coverage of 92.2% among participants who lived within 2 km of lake Chilwa. In Iraq, Lam et al[30] reported an overall two-dose OCV coverage of 87% in the targeted camps. Other studies have reported low coverage. A study conducted in Guinea by Luquero et al[31] reported a two-dose coverage of 75.5%. In a Mozambican survey, Semá Baltazar et al reported an overall two-dose coverage of 51.2% and 69.5% for at least one dose.

Our findings show a vaccine drop-out rate of 18.8%. This finding is contrary to the WHO guidelines which recommend that, to be protected, an individual needs to receive two doses of Euvichol-plus.[32] The WHO guidelines also recommend that there must be a minimum of 2 weeks delay between two doses. Two doses of Euvichol-plus provide protection against cholera at least for 3 years,

while one dose provides short-term protection.[33] Similarly, Bekolo *et al*[34] showed that patients who receive two doses of the OCV are 4.5 times less likely to develop severe disease than unvaccinated patients. Thus, individuals who receive only one dose do not get optimal protection from the vaccine. Future studies should focus on factors contributing to vaccine drop out after an individual has received the initial dose. Addressing these factors would increase coverage for both the first and second dose and maximise population benefits from the vaccine.

Finally, our findings show that majority of participants were females in the age range 5–14 and 15–35. The lowest coverage was among the older people aged above 65 years. These findings contradict those reported by Elias Chitio *et al*[35] which showed low vaccination coverage among Mozambican female respondents. The reason for this discrepancy is not clear; it could be due to sampling variation. Studies conducted on health service utilisation have reported high utilisation rates among females.[36] For example, Sialubanje *et al*[37] reported a low COVID-19 vaccine coverage among males in Zambia. The main reason for the vaccine hesitancy and low coverage were alcohol, payer, traditional remedies, limited understanding and misconceptions, adverse effects and limited information.

These findings are consistent with those reported in a post OCV campaign survey by Semá Baltazar *et al* in Mozambique which reported the highest coverage for the two doses among adult females and lowest among adult males. Our findings are also in line with those reported by Burnett *et al*[38] in their study on the oral cholera vaccination coverage after the first global stockpile deployment in Haiti in 2014. These authors reported a higher percentage of children aged 5–14 years who received both recommended doses than did those ≥15 years. Public Health interventions planning OCV campaigns should take into consideration these sex and age differences in vaccine coverages in order to achieve optimal vaccine coverage.

## Limitations

Potential limitations for this study should be acknowledged. First, although a random satellite-based sampling technique was used to select households, inclusion of all eligible participants found in a household into the study may have introduced selection bias. Second, our findings are based on a survey conducted following an OCV campaign in four urban subdistricts of Lusaka City. The generalisability of these findings to other settings within the country may be limited.

Despite these limitations, we believe this study has provided evidence on the actual coverage of the OCV campaign based on the survey and clearly shown that the reported administrative coverage was much higher than the actual one from the survey. These findings provide evidence on the importance of validating administrative vaccination campaign coverage with survey data based on the actual vaccinated population. Moreover, our findings

confirm the feasibility of conducting an OCV campaign and achieving high coverage during a cholera outbreak in an urban setting like Lusaka city.[39] They corroborate previous campaigns and surveys conducted in other countries as well as the WHO guideline which recommend complementing traditional cholera control and preventions strategies (provision of safe drinking water, adequate sanitation, disease surveillance, health promotion and case management) with OCV campaign during an outbreak.

## Conclusion

These findings show that the 2018 OCV campaign coverage reported using administrative data was much higher than the actual one based on the population that was vaccinated. Follow-up surveys are needed to validate administrative coverage estimates. Reliance on health facility data alone may mask low coverage and prevent measures to improve programming. The findings also confirm the feasibility of conducting an OCV campaign during a cholera outbreak in an urban setting. Future studies should focus on determining the long-term effectiveness of single and two doses in the population as well as their cost-effectiveness.

**Author affiliations**
[1]School of Public Health, Levy Mwanawasa Medical University, Lusaka, Zambia
[2]School of Medicine, The Copperbelt University, Kitwe, Zambia
[3]Disease Surveillance, World Health Organization, Lusaka, Zambia
[4]Surveillance and Disease Intelligence, Zambia National Public Health Institute, Lusaka, Zambia
[5]Public Health, Zambia Ministry of Health, Lusaka, Zambia
[6]Communication Information & Research, Zambia National Public Health Institute, Lusaka, Zambia
[7]World Health Organization, Lusaka, Zambia
[8]Epidemic Preparedness and Response, Zambia National Public Health Institute, Lusaka, Zambia

**Acknowledgements** We thank Lusaka District Health Management Team who provided field research assistants and study participants. The GTFCC supported the country with OCV stocks and operations funds.

**Contributors** All authors contributed substantially to the development of this manuscript. VMM, CS, NK, BBM, OC, AMN, WN, NK, LM, RH, HM and NS contributed to the conception of the study. NK, BBM, OC, NS, HM and LM supervised the data collection process. WN, CS and RH conducted the data analysis. CS, HM, OC and BBM drafted the manuscript; BBM and VMM edited it; CS revised the manuscript. All authors read, commented on and approved the final manuscript. CS is the guarantor for the manuscript.

**Funding** Ministry of Health and WHO (award/grant number: N/A).

**Competing interests** None declared.

**Patient and public involvement** Patients and/or the public were involved in the design, or conduct, or reporting, or dissemination plans of this research. Refer to the Methods section for further details.

**Patient consent for publication** Consent obtained directly from patient(s).

**Ethics approval** This study involves human participants and was approved by ERES Converge Committee (2019-09-001). Participants gave informed consent to participate in the study before taking part.

**Provenance and peer review** Not commissioned; externally peer reviewed.

**Data availability statement** Data are available on reasonable request.



of the author(s) and are not endorsed by BMJ. BMJ disclaims all liability and responsibility arising from any reliance placed on the content. Where the content includes any translated material, BMJ does not warrant the accuracy and reliability of the translations (including but not limited to local regulations, clinical guidelines, terminology, drug names and drug dosages), and is not responsible for any error and/or omissions arising from translation and adaptation or otherwise.

**ORCID iDs**
Cephas Sialubanje http://orcid.org/0000-0002-9077-1436
Albertina Moraes Ngomah http://orcid.org/0000-0002-3435-1170

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
