## [Reviewer comments · BMJ Open]

ARTICLE DETAILS

TITLE (PROVISIONAL)	Euvichol-Plus Vaccine Campaign Coverage during the 2017/2018 Cholera outbreak in Lusaka District, Zambia: A Cross Sectional Descriptive Study
AUTHORS	Mukonka, Victor M; Sialubanje, Cephas; Matapo, Belem; Chewe, Orbrie; Ngomah, Albertina; Ngosa, Willaim; Hamoonga, Raymond; Sinyange, Nyambe; Mzyece, Hannah; Mazyanga, Lucy; Bakayita, Nathan; Kapata, Nathan

VERSION 1 – REVIEW

REVIEWER	Orach, Christopher Makerere University, Community Health and Behavioral Sciences
REVIEW RETURNED	07-Mar-2023

GENERAL COMMENTS	Overall, the manuscript is satisfactorily written. However, 1) In the abstract, please improve the write up of the statistical analysis. Why is fishers exact test used to stratify OCV coverage, where there very few entries in some of the cells? 2) The objective of the paper on page 6 could be shortened. 2) Under sampling technique, the author could briefly state (range) of how many persons were sampled per household.
--

REVIEWER	Forsberg, Birger C. Karolinska Inst, public Health Sciences
REVIEW RETURNED	24-May-2023

GENERAL COMMENTS	Comments to the authors The article is well written and the study appears to have been conducted with adequate quality. The results generated from the study are in line with earlier findings and as such they do bring limited new knowledge to the research field, apart from providing data on the implementation of the cholera vaccine campaign in the districts where the study was conducted. Our position this is supported by the authors' own conclusion that "The generalizability of these findings to other settings within the country may be limited." Below follows some more detailed comments on the article. Abstract The methods did not specify the use of a logistic regression model, including the measures of effects (adjusted odds ratio with 95% confidence intervals). It is therefore pertinent that authors capture the information in the methods (which currently suggests a descriptive analysis). What is rationale for the age group classification? Is there any evidence to support the classification, which is presently too wide?
--

	Strengths and weaknesses Authors stated that conducting the survey shortly after the OCV campaign minimised recall bias; however, in the methods, they stated that the survey was conducted following the second round of the 2018 OCV campaign. The question is what is the interval between the second round of campaign and data collection for this study? This needs to be carefully considered in order to justify the stated strength. The use of ODK is now routine and may not deserve to be considered a strength. Introduction The spelling of the 'world health organization' in lines 29-30 should be corrected: World Health Organization; the same applies to the spelling of GTFCC in lines 8-8 on page 7. There is information on when the first OCV campaign commenced, but nothing about the timeline for the second campaign. While the objective suggests that the study primarily focuses on OCV coverage, the last sentence (lines 36-37) suggests that the coverage of OCV campaigns and its acceptability among the Zambian population were the focus of the study. Authors need to clarify this throughout the manuscript. Methods Authors reported that face-to-face interview was conducted in the local language. While this is a good practice, it would help the reader if the authors clarify whether or not the interview questions were administered by the research assistants or interpreted by the survey participants. This information has an implication for information bias if not explicitly described—see details in line 17 on page 9. Under quality assurance, authors stated that householders who were not present during the field were revisited to avoid selection bias. Please specify the number of visits here. Data processing and analysis Authors provided a detailed descriptive analysis of the data collected, which is in line with the study's title and specific objectives. However, in the abstract, odds ratios for vaccine acceptance were presented, which I explained appeared to have deviated from the study's objective. Authors need to provide the analytical details in the methods to justify the inclusion of the results. Ethical considerations Authors stated that interviews were conducted in the house away from the public interference to ensure privacy for participants. For a household with about 4 persons, how was privacy issue addressed? This section of the methods seems too wordy. Please consider making it more precise. Results A total of 1,325 of participants was estimated as adequate to address the research question(s). However, authors stated in the result chapter (lines 29-3- of page 11) that a total of 2,000 participants were recruited into the study, suggesting that the sample size was probably estimated after data collection. If this was not the case, authors need to justify the reason for sampling more participants than necessary. E.g., authors stated in line 34 on page 11 that "a total final sample of 1,691 respondents (84.5%) were included in the final analysis."
--	---

	Again, consider my previous comments on the introduction of risk factors when the methods did not describe the methodological steps. Discussion The discussion concludes that the findings from the survey are much in line with findings from other surveys in similar settings. It would be useful if the authors could highlight what new knowledge the study brings to the field of cholera control. A key finding in the study is that the reported vaccine coverage is considerably higher than the one found in the household survey. There is little discussion on the reasons for the possibly overestimated coverage from the routine reports. Also, the authors just assumes that the data from the survey are valid and correct. We feel there should be an unbiased short discussion on the possible reasons for the discrepancy between the two sets of data. References References to websites should contain information on what date the website was accessed.
--	---

VERSION 1 – AUTHOR RESPONSE

Reviewer 1

Overall, the manuscript is satisfactorily written. However, 1) in the abstract, please improve the write up of the statistical analysis. Why is fishers exact test used to stratify OCV coverage, where there very few entries in some of the cells? 2) the objective of the paper on page 6 could be shortened; 3) under sampling technique, the author could briefly state (range) of how many persons were sampled per household

Response 1: We appreciate the reviewer’s comments. We have now corrected abstract to read: descriptive statistics were computed to summarise respondents’ characteristics and OCV coverage per dose. Bivariate analysis (chi square test) was conducted to stratify OCV coverage according to age and sex for each round. All analyses were conducted in STATA ($p < 0.05$) (page 2)

Response 2: We appreciate the reviewer’s comment; we have now edited and shortened the objective of the study (page 5)

Response 3: We have now included the missing statement under sampling to read: 1 to 6 participants were sampled per households (page 6)

Reviewer 2

The article is well written and the study appears to have been conducted with adequate quality. The results generated from the study are in line with earlier findings and as such they do bring limited new knowledge to the research field, apart from providing data on the implementation of the cholera vaccine campaign in the districts where the study was conducted. Our position is supported by the authors’ own conclusion that “The generalizability of these findings to other settings within the country may be limited.”

Below follows some more detailed comments on the article.

Abstract

Query: The methods did not specify the use of a logistic regression model, including the measures of effects (adjusted odds ratio with 95% confidence intervals). It is therefore pertinent that authors capture the information in the methods (which currently suggests a descriptive analysis).

What is rationale for the age group classification? Is there any evidence to support the classification, which is presently too wide?

Response 1: We appreciate the reviewer's comment. We have now edited the abstract to include use of logistic regression (page 2)

Response 2: We appreciate the reviewer's comment on the rationale for the age group classification. We agree that the classification is too wide. Unfortunately, we are unable to edit the classification on the vaccine coverage; this is how the data was collected on using the pre-tested electronic questionnaire which was uploaded on the electronic tablet using ODK Version 1.12.2. We used the correct classification in table 1 on descriptive statistics (page 9).

Strengths and weaknesses

Query: Authors stated that conducting the survey shortly after the OCV campaign minimised recall bias; however, in the methods, they stated that the survey was conducted following the second round of the 2018 OCV campaign. The question is what is the interval between the second round of campaign and data collection for this study? This needs to be carefully considered in order to justify the stated strength.

The use of ODK is now routine and may not deserve to be considered a strength.

Response: We have explained that the data collection was conducted within a month following the second round of the OCV campaign (page 5).

Response: We appreciate the reviewer's comment on the use of ODK. We have now edited this strength to read: Use of a pre-tested electronic questionnaire enhanced data quality and ensured internal validity of the study (page 3)

Introduction

Query: The spelling of the 'world health organization' in lines 29-30 should be corrected: World Health Organization; the same applies to the spelling of GTFCC in lines 8-8 on page 7.

Response: We appreciate the reviewer's observation. We have made the suggested corrections (Page 4 and 5)

Query: There is information on when the first OCV campaign commenced, but nothing about the timeline for the second campaign.

Response: We appreciate the reviewer's comment. We have now provided information on the timeline for the second campaign (see page 5)

While the objective suggests that the study primarily focuses on OCV coverage, the last sentence (lines 36-37) suggests that the coverage of OCV campaigns and its acceptability among the Zambian population were the focus of the study. Authors need to clarify this throughout the manuscript.

Response: We appreciate this observation. The study focused on the coverage of the OCV campaigns; we have now corrected this throughout the manuscript

Methods

Query: Authors reported that face-to-face interview was conducted in the local language. While this is a good practice, it would help the reader if the authors clarify whether or not the interview questions were administered by the research assistants or interpreted by the survey participants. This information has an implication for information bias if not explicitly described—see details in line 17 on page 9.

Response: We have edited this section; it now reads: "Research assistants administered the questionnaire and read out the interview questions during the face-to-face interview which were conducted in the local language".

Query: Under quality assurance, authors stated that householders who were not present during the field were revisited to avoid selection bias. Please specify the number of visits here.

Response: We appreciate the comment. In the data collection section we mentioned that two visits were made if the householders were not available during the first visits. The household was skipped if respondents were not found or refused to participate during the second visit (page 7)

Data processing and analysis

Query: Authors provided a detailed descriptive analysis of the data collected, which is in line with the study's title and specific objectives. However, in the abstract, odds ratios for vaccine acceptance were presented, which I explained appeared to have deviated from the study's objective. Authors need to provide the analytical details in the methods to justify the inclusion of the results.

Response: We appreciate this observation. We have edited the abstract and results section accordingly and removed the odds ratios (page 2 and 15)

Ethical considerations

Query: Authors stated that interviews were conducted in the house away from the public interference to ensure privacy for participants. For a household with about 4 persons, how was privacy issue addressed?

This section of the methods seems too wordy. Please consider making it more precise

Response: We have deleted this section as it is a duplication with the one on ethical approval appearing at the end of the manuscript (see page 8 and 18)

Results

Query: A total of 1,325 of participants was estimated as adequate to address the research question(s). However, authors stated in the result chapter (lines 29-3- of page 11) that a total of 2,000 participants were recruited into the study, suggesting that the sample size was probably estimated after data collection. If this was not the case, authors need to justify the reason for sampling more participants than necessary. E.g., authors stated in line 34 on page 11 that "a total final sample of 1,691 respondents (84.5%) were included in the final analysis."

Again, consider my previous comments on the introduction of risk factors when the methods did not describe the methodological steps.

Response: The sample size was estimated before data collection. The 2000 seems to be an over sampling error which was discovered during data analysis (page 6).

Response: We have deleted the section on risk factors (both in the abstract and results section) as it is not in line with the objective of the study and not described in the methods (page 14)

Discussion

Query: The discussion concludes that the findings from the survey are much in line with findings from other surveys in similar settings. It would be useful if the authors could highlight what new knowledge the study brings to the field of cholera control.

A key finding in the study is that the reported vaccine coverage is considerably higher than the one found in the household survey.

There is little discussion on the reasons for the possibly overestimated coverage from the routine reports. Also, the authors just assumes that the data from the survey are valid and correct. We feel there should be an unbiased short discussion on the possible reasons for the discrepancy between the two sets of data.

Response: We appreciate this very important comment. We have edited the whole discussion to include the suggested information (page 14)

References

Query: References to websites should contain information on what date the website was accessed.

Response: We appreciate the observation and have edited the reference section accordingly (page 19)

VERSION 2 – REVIEW

REVIEWER	Forsberg, Birger C. Karolinska Inst, public Health Sciences
REVIEW RETURNED	05-Sep-2023
GENERAL COMMENTS	All comments considered and acted upon. Well done.